# Health-Related Physical Fitness in Adolescents from Spain, Estonia and Iceland: A Cross-Sectional, Quantitative Study

**DOI:** 10.3390/sports10120188

**Published:** 2022-11-22

**Authors:** Pablo Galan-Lopez, Thordis Gísladóttir, Maret Pihu, Antonio J. Sánchez-Oliver, Francis Ries, Raúl Domínguez

**Affiliations:** 1Department of Communication and Education, Universidad Loyola Andalucía, 41704 Dos Hermanas, Sevilla, Spain; 2Research Center for Sport and Health Sciences, School of Education, University of Iceland, 105 Reykjavík, Iceland; 3Institute of Sport Sciences and Physiotherapy, Faculty of Medicine, University of Tartu, 51005 Tartu, Estonia; 4Departamento de Motricidad Humana y Rendimiento Deportivo, Universidad de Sevilla, 41013 Sevilla, Sevilla, Spain; 5Physical Education and Sports, Faculty of Educational Sciences, University of Seville, 41013 Sevilla, Sevilla, Spain

**Keywords:** adolescence, habits, lifestyle, overweight, physical inactivity, sedentary

## Abstract

The benefits of physical fitness (PF) for health are well-known. Low PF significantly contributes to the prevalence of obesity in adolescents, with an increased risk of developing chronic diseases. The objectives of the present study were to explore the health-related PF components and body composition levels in adolescents in three European cities, and their differences. The present study is a cross-sectional, descriptive, and quantitative research effort with 1717 participants aged between 13–16 years (48% girls), enrolled in public and private secondary schools in Seville (Spain), Reykjavik (Iceland) and Tartu (Estonia). The ALPHA fitness battery test was used with the following tests: handgrip strength, standing broad jump, 4 × 10 m speed-agility, 20 m shuttle run, and anthropometric variables. Regarding body composition, differences were detected for city and gender in height (*p* < 0.001), weight (*p* < 0.001), body fat percentage (*p* < 0.001), and waist circumference (*p* < 0.001); but no differences were reported for BMI for both city (*p* = 0.150) and gender (*p* = 0.738). Similarly, concerning PF, it was detected statistically significant differences between cities and gender in handgrip strength (*p* < 0.001), jump test (*p* < 0.001), speed-agility test (*p* < 0.001), and cardiovascular endurance in both variables (*p* < 0.001). In total, 26.8% of the boys and 27.3% of the girls were categorized as overweight; 18.1% of the boys and 31.2% of the girls had an excessive percentage of fat mass; and 22.7% of the boys and 22.2% of the girls showed an excessive waist circumference. The participants from Seville presented the lowest results in PF tests. In contrast, Reykjavik, with the highest results in the endurance and speed-agility tests, and Tartu, with higher results in the manual grip strength and long jump tests, shared the highest results.

## 1. Introduction

The benefits of physical activity (PA) and physical fitness (PF) for physical and mental health are well known [1,2,3]. Numerous studies have shown the value of PF for the well-being of youth [4]. Low PA and PF are significant contributors to the prevalence of obesity and being overweight in adolescents, with an increased risk of developing chronic diseases [5]. The establishment of life habits occurs during adolescence, and psychological and physiological changes take place [6]. Therefore, it is vital to assess and monitor the PF levels of adolescents, as they are an essential indicator of their lifestyle. For example, lower cardiorespiratory fitness and muscular strength are stronger predictors of cardiovascular and all-cause mortality than other risk factors [7]. Studies have shown that a high level of PF in adolescence is fundamental for maintaining good health and well-being and is a preventive factor against non-communicable diseases [8]. Despite this knowledge, existing research indicates low levels of PF in adolescents [9]. Comparably, previous studies have reported that body composition has a strong association with PF [10] and adolescents with normal-ranged body composition levels show greater overall PF compared with overweight and obese adolescents, regardless of gender [11]. Further, a recent systematic review reported that vigorous PA was a strong predictor of improved body composition in youth, particularly for reducing body fat and increasing fitness [12], highlighting the importance of tracking body composition along with PF and PA during this critical period in youth’s life. However, it is important to highlight the value of adequate body composition measures. Obesity indicators are often based solely on body mass index (BMI), which provides information about excess body weight, but does not discriminate fat from lean body mass [13,14]. When considering the health and body composition among adolescents, abdominal fat has a significant impact on health and hence other forms of assessing obesity should take this parameter into account; thus, it is imperative to assess BMI with both measures of body fat percentages and waist circumference [15]. Moreover, high levels of a sedentary lifestyle have increased the prevalence of pathologies that affect today’s society. Regarding the benefits of physical activity, the literature is extensive. It is considered a vital tool for increasing and maintaining a good quality of life, especially in post-pandemic times. It is recommended that young people perform at least 60 min of moderate to vigorous exercise every day of the week. Physically inactive adolescents are likely to become the sedentary and/or obese adults of the future [16,17].

PF refers to the full range of health-related physical qualities, such as muscular strength, cardiorespiratory fitness, speed of movement–agility, flexibility, and body composition [18]. The assessment and monitoring of PF are crucial to identifying and establishing appropriate public health strategies for children and youth [19]. As there are more than 15 different test batteries to evaluate health-related fitness in children and adolescents [20], the need for a standardized test is critical. The European Commission funded a project in 2007 that aimed to provide a set of instruments for assessing different PA domains as well as health-related PF in a comparable way. The ALPHA Fitness Battery Test measures these key components of health-related PF with anthropometric variables (weight, height, BMI, fat percentage) and PF variables (cardiovascular endurance, upper body strength, lower body power, and speed/agility) [21].

The assessment and monitoring of PF and adequate body composition measurements among adolescents are vital to identifying and establishing appropriate public health strategies for healthy development amongst adolescents. Further, monitoring the health-related fitness of youth is important to increase the physical function capacity needed for everyday life and thus avoid obesity and disease risks among adolescents [22]. Therefore, the present study aimed to explore the health-related PF components and body composition levels in European adolescents with the ALPHA test battery in three cities from different regions in Europe. The secondary aim was to describe the difference in PF and body composition based on the gender and the country of the participants.

## 2. Materials and Methods

### 2.1. Participants

The present study is a cross-sectional, descriptive, and quantitative research with 1717 participants aged between 13 and 16 years (48% girls and 52% boys), enrolled in public and private secondary schools in Seville (Spain), Reykjavik (Iceland) and Tartu (Estonia) (see Table 1). Spanish participants represent a Mediterranean country, Icelandic subjects as a Nordic country, and the Estonian sample as a Baltic country. They were of interest in expressing different geographical and cultural countries. Distribution of participants according to gender and age is summarized in Table 1. Data collection took place between October and mid-November for Icelandic and Estonian participants and during the months of October to December for the Spanish adolescents. The study was approved by the Ethics and Research Committee of Andalusia (Spain, Ref.: 0310-N-17), the National Bioethics Committee of Iceland (Ref.: VSNb2017030026/03.01) and the Ethics Committee of the University of Tartu (Estonia, Ref.: 281/T-10).

In the current study, the subjects were female and male students aged 13 to 16 years old, who had previously presented the signed informed consent by their parents or guardians. They were only recruited under the condition that they regularly attended school and were taking part in the physical education classes without any type of cognitive, physical, or motor limitations.

### 2.2. Measures

The extended version of the ALPHA fitness battery test was used to assess PF and the participants’ body compositions.

Body composition elements (weight, height, body mass index, and fat percentage) and physical fitness variables were evaluated by the Extended Alpha fitness battery [21]. BMI was estimated from the ratio between body weight (kg) and the height-squared of the participants (m^2^). The waist perimeter was measured with a non-flexible tape measure (Seca 201, Seca, Hamburg, Germany) at an 0.1 cm accuracy level. Measurements were located at the narrowest point on the surface between the iliac crest and the last rib at the end of an exhalation. Due to the large number of participants and the limited time, skin fold measurements were omitted and a bioimpedance system (Tanita Inner Scan BF-689, Tanita, Tokyo, Japan) certified by the U.S. Food and Drug Administration was used instead. For the handgrip test a hand dynamometer with an adjustable fit (TKK 5401 Grip D, Takey, Tokyo, Japan) was employed to assess the maximum strength of upper extremities. A research assistant indicated the participants the accurate procedure and how to regulate the dynamometer individually to the size of their hands. When conducting the test, participants were encouraged to “grip as hard as they could”, and to maintain grip strength for at least 2 s. The average value of the two attempts for each arm was recorded. A long jump test was performed in standing and from a static position to analyze the participants’ lower body power. The boys and girls stood in the starting position and then performed a vigorous jump with both feet together, intending to go as far as possible. The distance recorded was between the starting position and the heel closest to the jump line. The best of the two jumps was recorded.

To assess the participants’ agility speed, a 4 × 10 m test was conducted. A demonstration showed how the subjects had to run four times with the highest possible speed; the 10 m separating the two lines delimited the test. All participant adolescents accomplished two series and the best time being recorded with a hand stopwatch. The cardiorespiratory fitness was measured by a shuttle run test covering the distance of 20 m and back, adjusting the running pace to the rhythm of an external acoustic signal [9]. Once 8.5 km/h was reached, the test speed was increased by 0.5 km/h per minute. Each subject’s participation ended when they did not reach the line at the same time as the acoustic signal sounded two consecutive times or when, due to fatigue, they had to drop out.

The ALPHA fitness tests and the measurements of the body composition variables were carried out during the participants’ physical education (PE) classes. The test battery was structured as a circuit, and the different tests and measures were implemented successively. The shuttle run test was completed on a different class day.

### 2.3. Statistical Analysis

The results of the different variables of body composition and health-related PF are presented as means (M) ± standard deviations (SD). To verify the normality of the variables, a Kolmogorov–Smirnoff test was performed, and Levene’s test checked the homogeneity of variances. The possible differences in body composition and the results from the PF tests were analyzed by a multivariate analysis of variance (MANOVA), including city (Seville, Reykjavik, and Tartu) and gender (boys and girls) as independent variables. The covariables were: age, height, weight, BMI, body fat percentage and waist circumference. Subsequently, a Bonferroni Post-Hoc test was completed. As a complementary measure, eta-squared ($\eta_{p}^{2}$) was calculated [23]. The statistical significance was set at *p* < 0.05, and the SPSS statistical package (version 18.0, SPSS Inc., Chicago, IL, USA) was used for all statistical analyses.

## 3. Results

### 3.1. Body Composition According to City and Gender

Regarding the height of the participants (Table 2), statistically significant differences for the variables of city (F = 114.840; *p* < 0.001; $\eta_{p}^{2}$ = 0.118) and gender (F = 221.705; *p* < 0.001; $\eta_{p}^{2}$ = 0.115) were found, with statistically lower values in Seville and Reykjavik in comparison to Tartu, both for boys (*p* < 0.001) and girls (*p* < 0.001). Concerning weight according to city (F = 30.480; *p* < 0.001; $\eta_{p}^{2}$ = 0.034) and gender, statistically significant differences were also observed, being higher in boys than in girls (F = 52.100; *p* < 0.001; $\eta_{p}^{2}$ = 0.030). However, unlike the variables of height and weight, when analyzing the differences in BMI, no statistically significant differences were observed neither for the city (F = 1.896; *p* = 0.150; $\eta_{p}^{2}$ = 0.002) nor for gender (F = 0.112; *p* = 0.738$;\;\eta_{p}^{2}$ < 0.001) (see Figure 1).

However, regarding the body fat percentage levels, differences were observed for gender, being higher in girls than in boys in the three cities analyzed (F = 648.200; *p* < 0.001; $\eta_{p}^{2}$ = 0.275). Statistically significant differences were also found between cities (F = 18.738; *p* < 0.001; $\eta_{p}^{2}$ = 0.021). Thus, boys from Tartu had a lower body fat percentage than those from Seville (14.47 ± 0.47 vs. 17.82 ± 0.33; *p* < 0.001) and Reykjavik (14.47 ± 0.47 vs. 17.37 ± 0.49; *p* < 0.001). On the other hand, in the case of the girls, higher values were only found in the girls from Seville than in those from Tartu (26.94 ± 0.53 vs. 25.10 ± 0.53; *p* = 0.010) (see Figure 2). In relation to waist circumference, statistically significant differences were also detected for gender, being higher in boys than in girls (F = 99.834; *p* < 0.001; $\eta_{p}^{2}$ = 0.055 ). For the three cities, statistically significant differences were found (F = 11.535; *p* < 0.001; $\eta_{p}^{2}$ = 0.013), with the values being higher for boys from Tartu compared to those from Reykjavik (*p* < 0.001) and Seville (*p* < 0.001).

### 3.2. PF According to City and Gender

In relation to hand grip strength (Table 3), statistically significant differences were observed for the city variable (F = 16.003; *p* < 0.001; $\eta_{p}^{2}$ = 0.018), with statistically lower values in Seville compared to Reykjavik (*p* < 0.01). In turn, the participants from Reykjavik also presented lower values than those from Tartu (*p* < 0.01). Concerning gender, statistically significant differences were found with higher performance for the boys (F = 39.517; *p* < 0.001; $\eta_{p}^{2}$ = 0.023). In the case of the boys, statistically higher values were observed in the participants from Tartu compared to those from Seville (*p* < 0.001) and Reykjavik (*p* = 0.001). The girls, on the contrary, showed statistically higher values in Tartu than in Seville (*p* < 0.001) and Reykjavik (*p* < 0.001), while the girls from Reykjavik presented a higher performance as compared to those from Seville (*p* = 0.011).

Concerning the jump test, statistically significant differences were found between the three cities (F = 141.888; *p* < 0.001; $\eta_{p}^{2}$ = 0.143). The participants from Seville had lower values than those from Reykjavik and Tartu, which, in turn, had higher records than those from Reykjavik. Regarding gender, statistically higher jump length values were also observed in boys compared to girls (F = 46.815; *p* < 0.001; $\eta_{p}^{2}$ = 0.027). In the case of the boys, it was observed that the participants from Tartu presented significantly higher values than those from Reykjavik (*p* = 0.001) and Seville (*p* < 0.001), the results of the participants from Reykjavik being higher than those from Seville (*p* < 0.001). In girls, lower values were observed in those from Seville compared to those from Reykjavik (*p* < 0.001) and Tartu (*p* < 0.001), while the girls from Tartu showed higher results than those from Reykjavik (*p* = 0.003).

Concerning the results of the speed-agility test, statistically significant differences were found between the three cities (F = 38.569; *p* < 0.001; $\eta_{p}^{2}$ = 0.043), where Seville showed lower levels than Reykjavik (*p* < 0.001) and Tartu (*p* < 0.001). According to gender, it was also observed that the performance was lower in boys than in girls (F = 8.675; *p* < 0.001; $\eta_{p}^{2}$ = 0.005). When analyzing the results of the boys from the participating cities, it was observed that the boys from Seville were slower than those from Reykjavik (*p* = 0.002). In this test, the girls from Seville showed lower performance levels than those from Reykjavik (*p* < 0.001) and Tartu (*p* < 0.001).

In relation to cardiovascular endurance, statistically significant differences were observed in the comparison between cities (F = 71.959; *p* < 0.001; $\eta_{p}^{2}$ = 0.078); the participants from Seville completed fewer stages than those from Reykjavik (*p* < 0.001) and Tartu (*p* < 0.001). Furthermore, subjects from Reykjavik completed more stages than those from Tartu (*p* = 0.001). Considering the gender of the students, the boys completed a more significant number of stages than girls (F = 9.827; *p* < 0.001; $\eta_{p}^{2}$ = 0.006). As for the boys, it was observed that those from the city of Seville completed a lower number of stages compared to those from Reykjavik (*p* < 0.001) and Tartu (*p* = 0.001). In turn, those from Reykjavik reached fewer stages than those from Tartu (*p* < 0.001). In the case of the girls, those from Seville completed fewer stages than those from Reykjavik (*p* < 0.001) and Tartu (*p* < 0.001).

## 4. Discussion

### 4.1. Body Composition According to City and Gender

Currently, there is great interest in the physiological differences between boys and girls that may affect the prevention, diagnosis and treatment of obesity and being overweight, in addition to their comorbidities [24]. The males have more lean mass, while the females have more body fat with the same BMI [25].

Regarding the body composition of the participants from the three cities, considering the results of the BMI of the sample, it can be observed that they are slightly higher than those obtained in studies in a similar population [26,27]. These results align with the increase and prevalence of overweight and current obesity in the adolescent population and the results in terms of waist circumference and fat percentage [28]. Furthermore, despite the widespread use of BMI as an indicator of adiposity in the population, its correlation with body fat is relatively poor [29]. Therefore, to correct this aspect, the BMI measurements have been complemented in the current study with those of percentage of body fat and waist circumference [30].

Concerning the significant differences between boys and girls, the literature reflects a higher percentage of girls than boys with excess adiposity in the participating populations [26]. Although this aspect needs to be studied further, a possible cause could be that girls practice less PA than boys and overall have adopted a more sedentary lifestyle [31]. On the other hand, the three hours of PE classes per week in Iceland compared to those in Seville and Tartu could explain the differences related to the participants’ body composition since it would imply a greater amount of PA per week.

Concerning the significant differences related to the elements of body composition among the participants from the three cities, these could be explained by a greater time of exposure to screens, a decrease in the levels of PA and an increase in sedentary attitudes typical of adolescence [32,33]. These behaviors favor a dangerous increase in body composition levels, a low level of muscle mass due to insufficient stimulus for optimal growth and, finally, support the development of physical illiteracy, originating the so-called “Pediatric Inactivity Triad” [34].

### 4.2. PF According to City and Gender

Having mentioned the consequence of a good level of health-related PF, the importance of its evaluation cannot be ignored since, again, an adequate level of PF is directly related to a good quality of life and is inversely related to early mortality [35].

The statistically significant differences in the test results obtained by boys in comparison to girls are something that is present in the scientific literature [30,36]. There are several possible explanations for the different results in the physical tests. Firstly, girls tend to be less physically active, which can directly influence the results of the tests [37]. Secondly, the greater participation of boys in PE classes and sports teams is also relevant and deserves to be considered for explaining this difference between boys and girls. For example, PE accounted for 11% of daily moderate-to-vigorous PA for children, while the percentage of total PA on days without PE class was much lower [38]. In addition, earlier studies have also indicated the relation between biological maturation, physical fitness and fat percentage in males [39].

On the other hand, only 34% of the Andalusian adolescents practice moderate-to-vigorous PA at least four times a week [31]. In comparison, this percentage is significantly higher (64.3%) among the adolescent population of Reykjavik [40]. Therefore, considering that a regular moderate-to-vigorous PA is related to better development and maintenance of basic physical abilities, this could explain the differences [41]. In addition, Spanish adolescents also showed lower PA levels than adolescents from Estonia [37], which could explain the differences in the lower results of the participants from Seville compared to those from Tartu.

In relation to the significant differences between the city of Tartu and the cities of Seville and Reykjavik, the participants from Tartu obtained better scores than those from Seville and Reykjavik on all the tests except the endurance and speed-agility test. In contrast, the adolescents from Reykjavik showed the best results. To explain the differences, the existing variations in the body composition of participants from the three cities must be considered. On the one hand, even without significant differences, the participants from Tartu showed the highest BMI. On the other hand, boys and girls showed the lowest body fat percentage. Therefore, it can be deduced that the percentage of muscle mass of the participants from Tartu is higher, which could be a determining factor in obtaining the best performance in handgrip strength and long jump tests. These tests require fast and explosive movements of short duration, demanding a higher percentage of muscle mass for optimal performance. These data are supported by several research studies that linked elements of body composition and performance in various fitness tests [38,42]. Concerning the better results of the participants from Reykjavik in the cardiovascular endurance test and the speed-agility test, the subjects from Seville and Tartu, even without presenting significant differences compared to the Reykjavik participants, had higher BMI values. Therefore, the lower performance in this test of the participants from these two cities could be due to their higher BMI since the endurance or cardiovascular fitness test and the speed-agility test involve propulsion or lifting of their own body weight [21].

The better result shown by the Estonian population in the strength tests (long jump and dynamometry) could be explained by the differences in the percentage of body fat compared to the participating population of Spain or Iceland since a higher BMI, and a better percentage of body fat could mean a higher percentage of muscle mass. This would coincide with a recent systematic review of temporal trends in adolescent dynamometry between 1967 and 2017, covering more than two million participants [43]. Furthermore, with regard to the possible association between organized sports participation and physical fitness, adolescents that take part in organized sports show better physical fitness than their counterparts. Adolescents that participated often in sport within sport clubs show both better physical fitness and mental condition compare to those who participated sometimes or never [44]. The regular participation in sports clubs is related mostly with the level of endurance and strength as the essential components of health-related fitness [45].

Finally, and regarding the effect of body composition (BMI, percentage of body fat and waist perimeter) on the performance of adolescents in PF tests, high values have a demonstrated effect on the performance in different tests of PF [46]. For this reason, and despite the absence of significant differences in BMI, the adolescent participants from Reykjavik showed the best results in the endurance test and the lowest in terms of BMI, which could be explained based on the aforementioned study. The present research calls on educational institutions and teachers to promote, develop and maintain optimal health-related physical fitness, as the literature has shown the vital role that such agents play in this development. [47].This study has several limitations. Firstly, our findings cannot be extended to the entire school population in the three countries where data was collected; however, the characteristics of the participants give us confidence that they represent a large proportion of the adolescent population in these cities. However, our findings can be used as valuable indicators that could form the basis for future research. Third, health and lifestyle depend on many other variables that have not been analyzed, such as sleep and nutrition. These variables would not only complement the study but would also help to improve the results presented and be able to draw better conclusions. Fourth, adolescence is a complex period, since there are differences between chronological and biological age that can affect aspects related to physical fitness. Finally, and despite having adolescent populations from three European countries belonging to different geographical regions, it was not possible to analyze variables related to cultural aspects that could determine patterns of physical activity practice. Nevertheless, our work could be the starting point for subsequent studies looking at socio-cultural aspects, and the health habits of the population studied.

## 5. Conclusions

Regarding the body composition of the participants, differentiated by sex, more than 20% of the boys and 22% of the girls had an excessive waist circumference. In addition, more than 27% of the girls and 26% of the boys were overweight according to their BMI. Finally, 18% of the male participants and 31% of the female participants had an excessive fat percentage. As for the results of the Alpha test battery, the student participants from Tartu and Reykjavik obtained the highest results in the cardiovascular endurance, speed-agility, lower body power (long jump) and upper body power (manual dynamometry) tests; the participants from Seville had the lowest results in all tests. All of the above highlights the importance of the results obtained, as monitoring and controlling the elements of health-related fitness and body composition at an early age will be a vital tool in the fight against non-communicable diseases.

## Figures and Tables

**Figure 1 sports-10-00188-f001:**
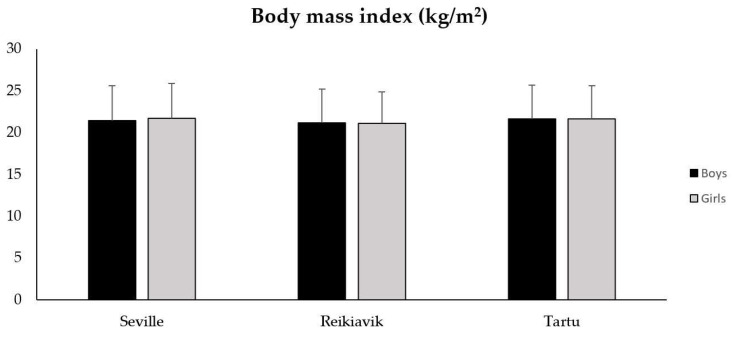
Body mass index (BMI) values in boys and girls from Seville, Reykjavik and Tartu. Data presented as M ± SD.

**Figure 2 sports-10-00188-f002:**
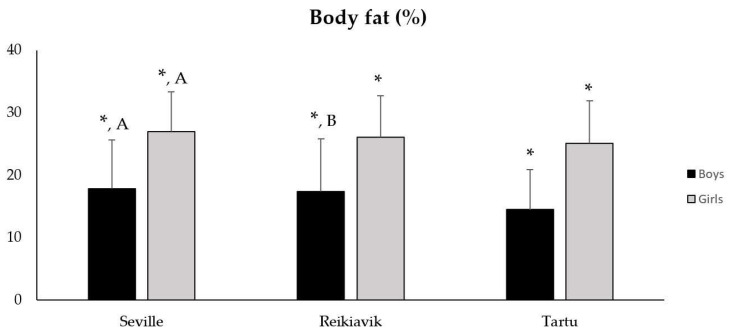
Body fat percentage values in boys and girls from Seville, Reykjavik and Tartu. Data presented as M ± SD; * Statistically significant differences between boys and girls from the same city; ^A^ Statistically significant differences between Seville and Tartu; ^B^ Statistically significant differences between Reykjavík and Tartu.

**Table 1 sports-10-00188-t001:** Distribution of the 1717 participants from Seville, Reykjavik and Tartu divided by gender.

City	Gender	13 Years	14 Years	15 Years	16 Years
Seville (n = 917)	Boys	113	131	115	99
	Girls	109	134	121	95
Reikiavik (n = 387)	Boys	42	51	47	69
	Girls	42	43	43	50
Tartu (n = 413)	Boys	50	62	48	73
	Girls	29	42	53	56
Total (n = 1717)	Boys	205	244	210	241
	Girls	180	219	217	201

**Table 2 sports-10-00188-t002:** Results of body composition variables in boys and girls from Seville, Reykjavik and Tartu.

Variable	Gender	Seville (*n* = 917)	Reykjavík (*n* = 387)	Tartu (*n* = 413)	*p*-Value City	*p*-Value Gender	*p*-Value City·Gender
Height (m)	Boys	1.65 ± 0.11 *	1.67 ± 0.11 *	1.74 ± 0.09 *	<0.001	<0.001	0.068
	Girls	1.58 ± 0.07 *	1.61 ±0.08 *	1.66 ± 0.07 *			
Weight (kg)	Boys	59.03 ± 0.62 *^, A^	59.20 ± 0.92 *^, B^	66.20 ± 0.87 *	<0.001	<0.001	0.272
	Girls	54.72 ± 0.62 *^, A^	55.10 ± 1.00 *^, B^	59.52 ± 0.99 *			
Waist Perimeter (cm)	Boys	74.37 ± 0.45 *^, A^	73.94 ± 0.67 *^, B^	77.82 ± 0.63 *	<0.001	<0.001	0.189
	Girls	69.68 ± 0.50 *	69.93 ± 0.72 *	71.43 ± 0.72 *			

Data presented as M ± SD; * Statistically significant differences between boys and girls from the same city; ^A^ Statistically significant differences between Seville and Tartu; ^B^ Statistically significant differences between Reykjavík and Tartu.

**Table 3 sports-10-00188-t003:** Results of the different tests of the Alpha-Fitness battery in boys and girls from Seville, Reykjavik and Tartu.

Variable	Gender	Seville (*n* = 917)	Reykjavík (*n* = 387)	Tartu (*n* = 413)	*p*-Value City	*p*-Value Gender	*p*-Value City·Gender
Handgrip Strength (kg)	Boys	27.07 ± 7.23 * ^A^	28.18 ± 7.90 * ^B^	32.87 ± 8.19 *	<0.001	<0.001	0.017
	Girls	21.72 ± 3.94 * ^A, C^	23.33 ± 4.58 * ^B^	25.67 ± 5.02 *			
Jump (m)	Boys	1.66 ± 0.32 * ^A, C^	1.83 ± 0.29 * ^B^	1.92 ± 0.31 *	<0.001	<0.001	0.001
	Girls	1.32 ± 0.25 * ^A, C^	1.59 ± 0.26 * ^B^	1.69 ± 0.24 *			
Speed (s)	Boys	11.91 ± 1.39 * ^B^	11.53 ± 1.59 *	11.68 ± 1.44 *	<0.001	<0.001	<0.001
	Girls	13.20 ± 1.34 * ^A, C^	12.12 ± 1.05 *	12.18 ± 1.08 *			
Endurance (stages)	Boys	6.09 ± 2.37 * ^A, C^	7.52 ± 2.62 *, ^B^	6.72 ± 2.46 *	<0.001	<0.001	0.043
	Girls	4.22 ± 1.40 * ^A, C^	5.80 ± 2.04 *	5.48 ± 1.73 *			

Data presented as M ± SD; * Statistically significant differences between boys and girls from the same city; ^A^ Statistically significant differences between Seville and Tartu; ^B^ Statistically significant differences between Reykjavik and Tartu; ^C^ Statistically significant differences between Seville and Reykjavik.

## Data Availability

Data originated during the research project is available on request from the corresponding author (sanchezoliver@us.es).

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
