# Peer review of "Health-Related Physical Fitness in Adolescents from Spain, Estonia and Iceland: A Cross-Sectional, Quantitative Study"

_sports, 2022, doi:10.3390/sports10120188_

Round 1
Reviewer 1 Report
Overview
In the present study, the authors aimed to explore the health-related Physical Fitness components and body composition levels in European adolescents with the ALPHA test battery in three cities (Seville, Reykjavik and Tartu) from different regions in Europe (Spain, Iceland and Estonia, respectively). Furthermore, they aimed to describe the difference in Physical Fitness and body composition based on gender and country of participants.
The study is important and well-written. However, there are some issues to address in the methods section before making the manuscript publishable.
Specific comments
-Please standardize the name of the city of Seville in the manuscript. Sometimes it is spelled Sivilla (Spanish) and others Seville (English).
Materials & Methods
Participants
-Please add the period in which the study was carried out.
-Informed consent? Were the participants made to sign? If not, this study is not publishable.
-Add some descriptive statistics, i.e. the number of participants per country.
Measures
This section should be completely rewritten and expanded to better describe each individual test; how were they performed? How was body composition calculated? The equation used? Add the manufacturer and model of the handgrip, skinfold/BIA, balance and statimetre. How were the test times recorded? With a hand stopwatch or with photocells? The manufacturer and model of the chronometer or photocells? How many times have the tests been repeated? What value was recorded (the best or the average value)? In what order? In which and how many days? What was the reliability of the measures (ICC)?
Tables 1 and 2
-Replace commas with periods before decimals.
-Add the number of participants for each city. For example, Sevilla (= 400).
Author Response
Overview
In the present study, the authors aimed to explore the health-related Physical Fitness components and body composition levels in European adolescents with the ALPHA test battery in three cities (Seville, Reykjavik and Tartu) from different regions in Europe (Spain, Iceland and Estonia, respectively). Furthermore, they aimed to describe the difference in Physical Fitness and body composition based on gender and country of participants.
The study is important and well-written. However, there are some issues to address in the methods section before making the manuscript publishable.
Dear reviewer,
Thank you very much for your carefully revision and constructive comments. We appreciate your suggestions that we have considered all of them. After this revision, the quality of the manuscript has enhanced.
Specific comments
-Please standardize the name of the city of Seville in the manuscript. Sometimes it is spelled Sivilla (Spanish) and others Seville (English).
It has been revised and standardized the term Seville along the manuscript.
Materials & Methods
Participants
-Please add the period in which the study was carried out.
Data collection period has been specified.
-Informed consent? Were the participants made to sign? If not, this study is not publishable.
All participants provided an informed consent document. You’ll find extra information at the end of the document, at informed consent statement section.
-Add some descriptive statistics, i.e. the number of participants per country.
Extra information have been added in table 1
Measures
This section should be completely rewritten and expanded to better describe each individual test; how were they performed? How was body composition calculated? The equation used? Add the manufacturer and model of the handgrip, skinfold/BIA, balance and statimetre. How were the test times recorded? With a hand stopwatch or with photocells? The manufacturer and model of the chronometer or photocells? How many times have the tests been repeated? What value was recorded (the best or the average value)? In what order? In which and how many days? What was the reliability of the measures (ICC)?
Dear reviewer, following your instructions,measures section has been rewritten in order to provide more information to the reader.
Tables 1 and 2
-Replace commas with periods before decimals.
Done
-Add the number of participants for each city. For example, Sevilla (= 400).
More information about the participantes of each city has been provided in table 1
Reviewer 2 Report
Dear authors,
Your submitted paper "Health-related Physical Fitness in European Adolescents: a Cross-Sectional, Quantitative Study” is beneficial for this field of science
The paper was well-written overall and provided some great insight into a relevant and important topic. However the authors should endeavor to improve the quality of the scientific basis of the study.
Method - The methodology seems appropriate. I think the participants section could be expanded
overall, a very well conceptualized, constructed, and written paper.
Line 84 (Materials and Methods): Why were these three cities chosen for the research?
Line 85 (Participants): How was the universe and sample selection made in the study?
Line 85: (On the selection of participants) Provide information about subjects' inclusion or exclusion from the study. Please explain
Line 85 (Participants): give more details on the number of participants (schools by age and gender)
Line 93 (Measures): Which instruments were used for the measurements? Please explain
Line 93 (measures): Body Fat (%) measured with skinfold?
Line 116: (Body composition and PF according to city and gender) effect size data add
Discussion - This section was nicely constructed and clearly outlined how the findings were connected with previous literature and how the findings can be used practically.
Author Response
Dear authors,
Your submitted paper "Health-related Physical Fitness in European Adolescents: a Cross-Sectional, Quantitative Study” is beneficial for this field of science
The paper was well-written overall and provided some great insight into a relevant and important topic. However the authors should endeavor to improve the quality of the scientific basis of the study.
Dear reviewer,
Thank you very much for your carefully revision and constructive comments. We appreciate your suggestions that we have considered all of them. After this revision, the quality of the manuscript has enhanced.
Method - The methodology seems appropriate. I think the participants section could be expanded overall, a very well conceptualized, constructed, and written paper.
Line 84 (Materials and Methods): Why were these three cities chosen for the research?
Explanation about the reasons of these three cities have been provided.
Line 85 (Participants): How was the universe and sample selection made in the study?
The sample selection was done by selected schools that represents students with different socioeconomical status and all schools in Spain, Estonia and Iceland were municipality schools.
Line 85: (On the selection of participants) Provide information about subjects' inclusion or exclusion from the study. Please explain
Inclusion criteria has been detailed.
Line 85 (Participants): give more details on the number of participants (schools by age and gender)
Extra information have been added in table 1
Line 93 (Measures): Which instruments were used for the measurements? Please explain
Dear reviewer, following your instructions,measures section has been rewritten in order to provide more information to the reader.
Line 93 (measures): Body Fat (%) measured with skinfold?
Dear reviewer, following your instructions,measures section has been rewritten in order to provide more information to the reader.
Line 116: (Body composition and PF according to city and gender) effect size data add
It has been added effect size (ES) as body composition as PF variables.
Discussion - This section was nicely constructed and clearly outlined how the findings were connected with previous literature and how the findings can be used practically.
Thank you very much for your positive feedback.
Reviewer 3 Report
Dear Authors
the following revisions are needed to be done or logical justification should be made. the strength of your work refers to studying 1717 participants which is valuable and highly appreciated.
1- You’ve used the term "European children, while you have studied just three cities in Europe. Is it possible to generalize it to Europe? if not, please revise your research title.
2- specify participants information (e.g., age; physical characteristics....).
3- check key words-based Mesh standard. Better no included those key wors that are included in research title and also abstract (if possible).
4- as you know, titles such as "Health-related Physical Fitness in adolescents” have been conducted for several times worldwide. you are recommended to focus more on research necessity in Introduction. one good justification would be related to increasing rate of obesity in this critical period which influence the upcoming years (adulthood). in this regard, use the following reference:
Abdelkarim, Osama, et al. "Prevalence of underweight and overweight and its association with physical fitness in Egyptian schoolchildren." International journal of environmental research and public health 17.1 (2020): 75.
5- it’s clear that components of Health-related Physical Fitness include the following ones:
"Cardiovascular endurance; Muscular strength; Muscular endurance; Flexibility; Body composition" in this regard, you have used the "Health-related Physical Fitness" I your research title, however, cardiovascular fitness is NOT considered in The extended version of the ALPHA fitness. how do you justify that?
6- in result section of abstract, report the results using P-values.
7- there are some factors affecting the adolescents’ physiological functions, especially sleep and nutrition. you can mention them for future studies and can be considered as your study limitation.
8-. Sampling recruitment better be depicted by flow chart.
9- better to show some parts of the results by figures. Additionally, all p-Values should be reported by exact values instead of e.g., p < 0.001.
10- what are the physiological and developmental mechanisms involve in obtained results?
11- in your discussion section, better talk about The Association Between Sports Participation and Physical Fitness. you can use the following reference for that:
Tahira, Shazia. "The Association Between Sports Participation and Physical Fitness." International Journal of Sport Studies for Health 4.2 (2021).
12-Since one of the effective factors for promoting physical activity in adolescents and improving their PF is achieved in schools and most of encouragement policies are done at schools by peers and sport teachers, you should recommend it and can use the following reference for that.
Sayar, Insaf, et al. "Impacts of Teachers’ Pedagogical Approach on the Inclusion of Adolescents with Exceed Weight into Physical Education and Sports in Emirate of Ajman/United Arab Emirates." Annals of Applied Sport Science 10.2 (2022): 0-0.
13- it seems that more study limitations can be mentioned in your study. Be more specific.
Author Response
Dear Authors, the following revisions are needed to be done or logical justification should be made. the strength of your work refers to studying 1717 participants which is valuable and highly appreciated.
Dear reviewer,
Thank you very much for your carefully revision and constructive comments. We appreciate your suggestions that we have considered all of them. After this revision, the quality of the manuscript has enhanced.
1- You’ve used the term "European children, while you have studied just three cities in Europe. Is it possible to generalize it to Europe? if not, please revise your research title.
Dear reviewer, following your indications, the tittle has been modified.
2- specify participants information (e.g., age; physical characteristics....).
Age, sex and inclusion criteria details have been included in section 2.1.
3- check key words-based Mesh standard. Better no included those key wors that are included in research title and also abstract (if possible).
Dear reviewer, following your indications, key words have been modified.
4- as you know, titles such as "Health-related Physical Fitness in adolescents” have been conducted for several times worldwide. you are recommended to focus more on research necessity in Introduction. one good justification would be related to increasing rate of obesity in this critical period which influence the upcoming years (adulthood). in this regard, use the following reference:
Dear reviewer, we have included new statements at the introduction section and used the proposed reference.
5- it’s clear that components of Health-related Physical Fitness include the following ones:
"Cardiovascular endurance; Muscular strength; Muscular endurance; Flexibility; Body composition" in this regard, you have used the "Health-related Physical Fitness" I your research title, however, cardiovascular fitness is NOT considered in The extended version of the ALPHA fitness. how do you justify that?
Dear reviewer, you will find the test manual for the alpha fitness battery in the following link à https://www.ugr.es/~cts262/ES/documents/ALPHA-FitnessTestManualforChildren-Adolescents.pdf . On page six you will find detailed information about the tests that make up the extended version of the battery. Among them is the 20m shuttle run test that asses cardiorespiratory fitness.
Cardiovascular fitness is one of the important component of health-related fitness and it is considered also in research title and in research.
6- in result section of abstract, report the results using P-values.
It has been reported all the p-value of ANOVA.
7- there are some factors affecting the adolescents’ physiological functions, especially sleep and nutrition. you can mention them for future studies and can be considered as your study limitation.
Dear reviewer, limitations sections has been modified following your comments.
8- Sampling recruitment better be depicted by flow chart.
Extra information about sample has been included in table 1 and in participants section.
9- better to show some parts of the results by figures. Additionally, all p-Values should be reported by exact values instead of e.g., p < 0.001.
It has been revised and specified all p-value including 3 decimals. In addition, it has been added two figures that reflex important findings, differences in body fat, but not in BMI
10- what are the physiological and developmental mechanisms involve in obtained results?
Relatives aspects to your comments have been added to de discussion.Earlier studies have indicated also to the relation between biological maturation, physical fitness and fat percentage in males.
11- in your discussion section, better talk about The Association Between Sports Participation and Physical Fitness. you can use the following reference for that:
Tahira, Shazia. "The Association Between Sports Participation and Physical Fitness." International Journal of Sport Studies for Health 4.2 (2021).
Thank you very much for your advice and valuable comments, discussion section has been modified, including your proposed reference.
12- Since one of the effective factors for promoting physical activity in adolescents and improving their PF is achieved in schools and most of encouragement policies are done at schools by peers and sport teachers, you should recommend it and can use the following reference for that.
Dear reviewer, we have modified de final section of our manuscript
13- it seems that more study limitations can be mentioned in your study. Be more specific.
Dear reviewer, limitations sections has been modified following your comments.
Round 2
Reviewer 1 Report
Thank you for your replies.
Reviewer 3 Report
Accepted